# Study on the Behavior of the Water Temperature Inversion Layer in the Northern East China Sea

**Seong Hyeon Kim** [1] , **Bok Kyoung Choi** [2] **and Eung Kim** [1],*

[1]  Maritime Security and Safety Research Center, Korea Institute of Ocean Science and Technology, Busan 49111, Korea; kenta81@kiost.ac.kr

[2]  Marine Domain Management Research Division, Korea Institute of Ocean Science and Technology, Busan 49111, Korea; bkchoi@kiost.ac.kr

*  Correspondence: eung@kiost.ac.kr

**Abstract:** To investigate the behavioral characteristics of the water temperature inversion layer (TIL), we used data (KODC) from areas in the Northern East China Sea from 1995 to 2016. Water temperature and salinity surveys were conducted 8820 casts over 22 years. Of these, 1589 water temperature inversion layers were found, and the probability of occurrence was 18.0%. In the Gageo island, probability of TIL occurrence in winter was 25 times higher than in summer. On the other hand, in the south of Jeju Island, summer values were 3.7 times higher than winter values. A T–S diagram analysis shows the components of the water temperature inversion layers. Yellow Sea Cold Water was mainly found in the winter, while Jeju Warm Currents and Tsushima Warm Currents were found in summer. The correlation between the probability of the occurrence of a monthly water temperature inversion layer and the amount of seawater volume transported into the study area was analyzed. The correlation coefficient was higher than $r = 0.8$ in parts of southern Jeju Island. On the other hand, the correlation coefficient was $r = −0.6$ in the Gageo Island. The spatial correlation index for the seawater volume transport and the water temperature inversion layer is presented.

**Keywords:** water temperature inversion layer; Northern East China Sea; Tsushima Warm Current; volume transport; Jeju Island; Yellow Sea Cold Water; Jeju Warm Current

## 1. Introduction

In the ocean, in general, the water temperature gradually decreases as the water depth increases. However, at some depths, the water temperature of the middle and lower layers sometimes rises above that of the upper layer, which is called a water temperature inversion (TI). The causes of this water temperature inversion layer (TIL) may be due to: (1) Near heat loss from the sea surface [1], (2) wind-driven current moving from a cold water area to a warm water area [2], and (3) mixing process in the front area [3]. A TIL mainly occurs in coastal upwelling areas [4] and in polar fronts [3], where different water masses meet, and complex marine physical characteristics, such as water temperature fronts, are found near the boundaries of these areas [5].

According to [6], the water temperature inversion in the eastern sea of Jeju Island is due to the penetration of the Tsushima Warm Current (TWC), which has a lower water temperature and lower salinity compared to the Korea South Coastal Current (KSCC), which has higher water temperature and higher salinity. The study in [3] showed that winter and summer water TIL phenomena in the Yellow Sea are seven times more common in winter than in summer. The work in [7] reported that, in the South Sea of Korea, TIL occurred six times more frequently in winter than in summer during 1965 to 1979. According to the authors in [8], who studied the TIL phenomenon that occurs in the South Sea of Korea, different water masses (KSCC and TWC) form a front in the South Sea of Korea as

they expand and decrease with each season; the authors reported that the TIL occurs mainly in the front sea area. In addition, the study in [9], who studied the entrance of the Yellow Sea, the entrance of the Yellow Sea is the area where Korean Coastal Water meets JWC. In winter, a TIL occurs when cold Korean Coastal Water is present in the upper layer and warm JWC is present in the lower layer. According to the study in [1] (a statistical analysis of the TILs occurring in the East China Sea and the Yellow Sea), a TIL occurs near Jeju island where Korea Coastal Water, Yellow Sea Warm Current, and TWC meet below 35° N. In this area, the TILs are deep and strong. The authors in [10] reported that, based on the results of modeling in the south area of the Yellow Sea in winter, TILs occur due to Ekman transport by the north wind.

Therefore, a TIL is generated when different water masses meet, which can be considered the front area. Some TILs have a spatial change in water temperature and salinity because of high-temperature high-salinity water penetrating into the lower layer, which may affect the benthic ecosystem. In addition, most of the TILs are generated by the water temperature front and can be a major indicator for the distribution of fronts with various effects on marine life. The TIL is a phenomenon that occurs when water masses with different densities in their upper and lower layers meet. Most of the TILs that occur in the South Sea of Korea and the Yellow Sea occur when Korea Coastal Water, Yellow Sea Warm Water, JWC, or TWC meet.

Among the water masses affecting the Northern East China Sea (NECS), Yellow Sea Cold Water (YSCW), TWC, Changjiang Diluted Water (CDW), JWC, and Kuroshio Branch Current (KBC) are representative. CDW flows onto the surface of Jeju Island offshore and is classified with a water temperature up to 23 °C and salinity less than 31 psu [11]. JWC has a water temperature up to 12 °C and salinity ranging from 33.5 to 34.0 psu [12]. In addition, TWC is diverted from the east side of Jeju Island through Tsushima to the East Sea of Korea [13]; it has a water temperature up to 14.0 °C and salinity up to 34.1 psu [14]. The current flowing through the right side of Jeju Island is called the Kuroshio Branch Current (KBC), which is distinguished from the TWC and has a salinity value of 34.0 psu or more [12]. However, KBC and TWC showed only a 0.1 psu difference in their salinity values and were judged to have similar characteristics. In this paper, the water mass range of TWC, including KBC in TWC, featured a water temperature above 14.0 °C and salinity above 34.0 psu.

The classification criteria of various researchers for YSCW are different. The authors in [15] classified seawater with salinity ranging from 32.0 to 32.5 psu and water temperature below 10 °C, which is formed during the winter at the bottom of the central Yellow Sea and appears from spring until autumn. The study in [11] classified water temperatures below 14.5 °C and salinity below 33.7 psu. The authors in [16] collected 40 years of data on the Yellow Sea and the East China Sea and classified them into water temperatures below 10.0 °C and salinity below 33.2 psu. The authors of [17] classified their data into water temperature of 13.2 °C or lower and salinity of 32.6 to 33.7 psu. The authors in [18] classified seawater with a water temperature below 12 °C and salinity below 33.5 psu. There is no consistent definition of YSCW. In this study, to distinguish YSCW from JWC, the water temperature was set to 14.5 °C or less with 32.0 to 33.5 psu salinity. The range of the water mass is shown in Table 1. This range was used to distinguish the mass in the TIL.

**Table 1.** Range of water temperature and salinity values by water mass in the NECS [1].

| Parameter | CDW [2] | YSCW [3] | JWC [4] | TWC [5] (KBC [6]) |
|---|---|---|---|---|
| Temperature (°C) | 23.0 ≤ T | 5.0 ≤ T ≤ 14.5 | 12.0 ≤ T | 14.0 ≤ T |
| Salinity (psu) | 31.0 ≥ S | 32.0 ≤ S ≤ 33.5 | 33.5 ≤ S ≤ 34.0 | 34.0 ≤ S |

[1] NECS; Northern East China Sea [2] CDW; Changjiang Diluted Water, [3] YSCW; Yellow Sea Cold Water, [4] JWC; Jeju Warm Current, [5] TWC; Tsushima Warm Current, [6] KBC; Kuroshio Branch Current.

We studied the characteristics of the TIL near Jeju Island, where various currents coexist. For this study, we analyzed the monthly spatiotemporal distribution of the TIL phenomena, the water

mass characteristics of each TIL, and the correlation between monthly TIL occurrence and seawater volume transport.

## 2. Data and Methods

### 2.1. Observation Field Data

In order to understand the frequency of the annual and seasonal occurrence of TIL in the East China Sea, selected marine observational data from the National Institute of Fisheries Science were used at the Korea Oceanographic Data Center (KODC, http://www.nifs.go.kr/kodc/index.kodc). We used all CTD data (water temperature and salinity) along the 203, 204, 311, 312, 313, 314, 315, 316, 317, and 400 transects from 1995 to 2016 provided by KODC. We used 8820 casts for CTD data corresponding to the sea area around Jeju Island and the Yellow Sea (Figure 1). The 203, 204, 400, and 311 to 314 transects (•) were observed annually in February, April, June, August, October, and December. The 315 to 317 transects (◆) was measured annually in February, May, August, and November. The number of surveys were February 1688, April 1034, May 654, June 1034, August 1688, October 1034, November 654, and December 1034. The measuring cycle is different. In the case of the 317 transect, observation began in 2000, so the difference in observation period should be taken into account.

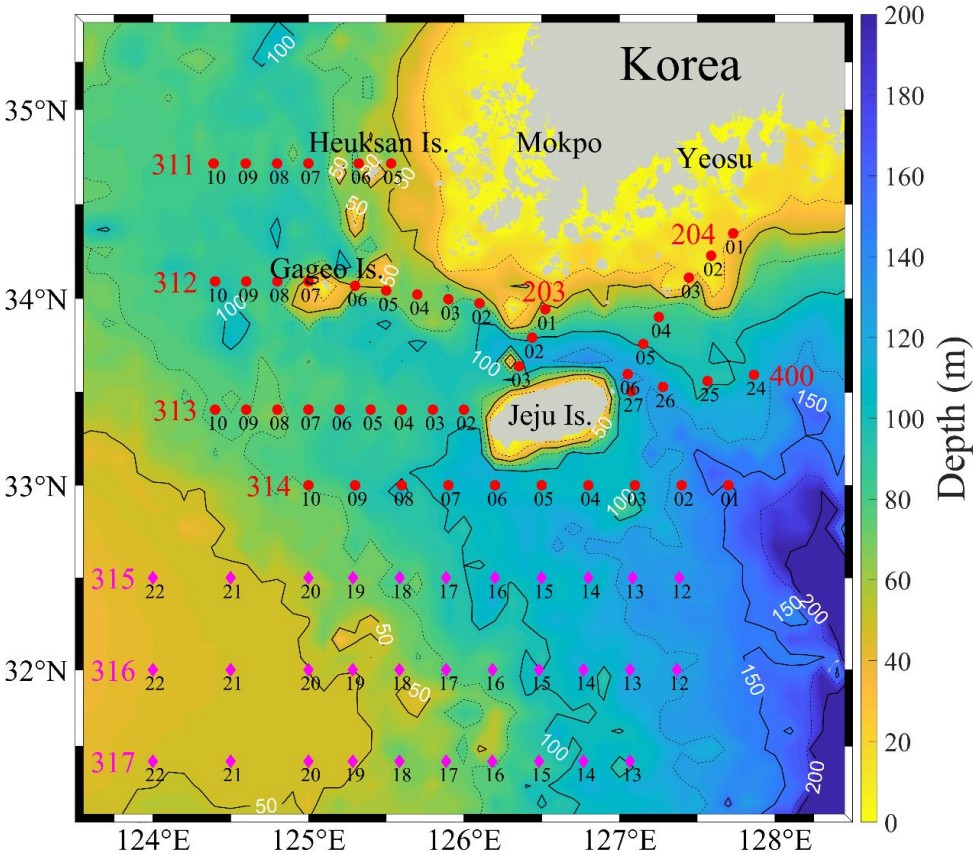

**Figure 1.** Station of the water temperature inversion layer (TIL) behavior analysis in the Northern East China Sea Island during 1995–2016. Red stations were observed in February, April, June, August, October, and December. Magenta stations were observed in February, May, August, and November. Black number is station of observation transect. The black line (dot line and white number) is bathymetry depth.

Occasionally, observations occurred at certain times due to bad sea weather. For example, April is a regular observation period, but was observed in early May. In the case of some discrepancies in the measurement times, the data were rearranged according to the data for April, which is a regular

observation period, to ensure uniformity among the long-term observational data. KODC provides data at standard depths of 0, 10, 20, 30, 50, 75, and 100 m. In order to analyze the characteristics of TIL, the interpolation was performed at a 1 m depth using the interpolation method ('griddedInterpolant.m' function provided by MATLAB).

### 2.2. Calculation of TIL Profile

A TIL is defined as when the lower water temperature is higher than the upper water temperature (Figure 2a). The water temperature and depth of the starting point of a TIL are expressed as $T_0$ and $D_0$, respectively, and the maximum water temperature of a TIL is determined as $T_1$ and $\Delta D$ as the depth $D_1$. TIL was defined when $\Delta T$ is more than 0.1 °C and $\Delta D$ is more than depth 10 m (Figure 2b). In order to analyze the behavior of a TIL, the probability of its occurrence was calculated, as in Equation (1).

$$P_{Mi} = \left(\frac{O_{Mi}}{N_{Mi}}\right) \times 100 \ (\%) \tag{1}$$

where $P_M$ is the probability of the occurrence of a monthly TIL, $N_M$ is the total number of observation stations per month, $O_M$ is the number of monthly occurrences of TIL, and $_i$ is the observation station. To identify the water mass affecting TIL in winter (February) and summer (August), the total water temperature and salinity profiles and profiles in TIL were analyzed in the T–S diagram.

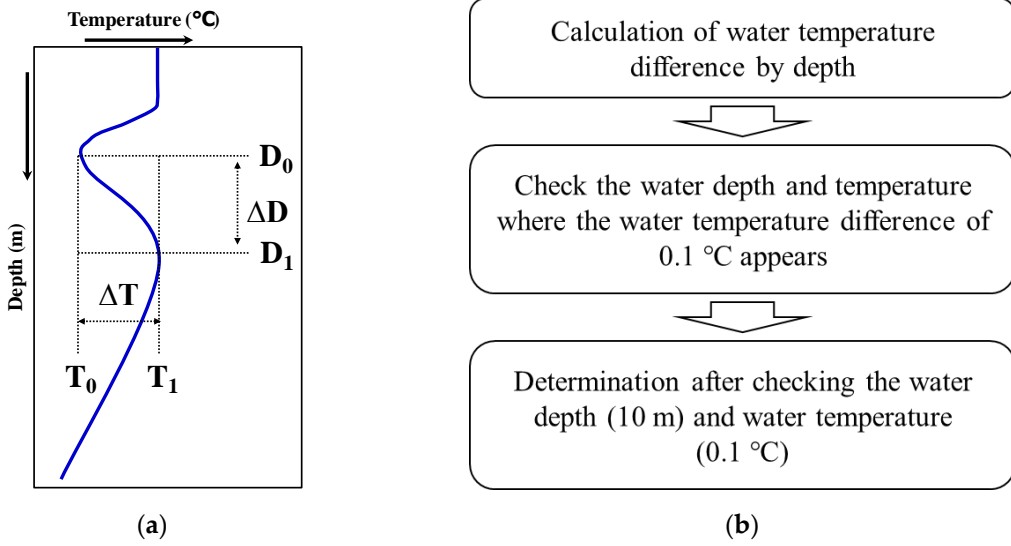

(**a**)    (**b**)

**Figure 2.** A schematic diagram showing of TIL (**a**). $T_0$ and $D_0$ is TIL starting point and $T_1$ and $D_1$ is maximum water temperature of a TIL. The flow chart for determination of water temperature inversion layer (**b**).

### 2.3. Calculation of Seawater Volume Transport

The relationship between the generation of TIL and the seawater transport influx of the NECS was examined. There is insufficient data on the annual and seasonal transport influx in this area. Transport estimates were based on the GLORYS12 data from the reanalysis model data provided by the Copernicus Marine environment monitoring service in Europe (http://marine.copernicus.eu/services-portfolio/access-to-products/). CLORYS12 is a model that reanalyzes the global current, water temperature, and salinity from January 1993 to December 2017 (http://marine.copernicus.eu/services-portfolio/access-to-products/?option=comcsw&view=details&productid=GLOBAL_REANALYSIS_PHY_001_030). The reanalyzed data are processed by the assimilation of field observations. The horizontal resolution is 1/12° (5′) and the depth resolution is divided at 50 layers from 0.5 to 5500 m. The depth interval increases linearly from 1 to 450 m. In this study, 30 layers were used up to 380 m depth. CLORYS12 reanalyzes the global data

and provides daily and monthly averaged data. In this study, monthly current depth data were extracted for 1995 to 2016. From the extracted data, the volume transport was calculated by using the current data for each depth at latitude 31° N and longitude 123–128° E. The reason for choosing the latitude 31° N is that it incorporates most of the currents that flow into the study area. The volume transport was calculated only for the northward component, because of the most currents flowing into the study area were northward. The volume transport calculated Sverdurp (Sv = $10^6$ m$^3$/s) that it was calculated by interpolating longitude and depth to 1 m.

## 3. Results

### 3.1. Monthly Occurrence of TILs

The monthly probability of an occurrence of a TIL at each station from 1995 to 2016 was analyzed. The measurement month for each observation transect is different. Therefore, there may be regional errors in the average monthly occurrence probability (Figure 3). In February, more than 70% of the maximum value was observed at the 312 transect, and most of the winter TILs occurred at the 312 transect. The most frequent occurrence was at the 312 transect in April, and up to 30% at some stations on the 314 transect. In May, only the southern part of Jeju Island was surveyed, and the probability of a TIL was higher than 40%. In June, the TIL probability was nearly 30% at the 314 transect south of Jeju Island. In comparison, the 312 transect was less than 10%. In August, more than 40% of the southern parts and nearby Jeju Island experienced a TIL, and less than 10% of the 312 transect experienced a TIL. In October, the occurrence was high at 30–40% in the transects (204, 313, 314, 400) near Jeju Island and was distributed below 10% on the 312 transect. In December, the trend increased to 25.3% from the 312 transect, which ranged from 2.5% to 9.6% from June to October. On the other hand, the probability of occurrence decreased at the 314 transect. When analyzing the characteristics of TILs by transects (Table 2), the probability of monthly TIL occurrence for each transect showed a large difference. In February and April, the probability of the appearance of a TIL was highest in the 312 transect, and in August, it was the highest in the 314 to 317 transects south of Jeju Island.

**Table 2.** The distribution of occurrences of TIL in the NECS during 1995 to 2016.

| Parameter | Total | Feb. | Apr. | May | Jun. | Aug. | Oct. | Nov. | Dec. |
|---|---|---|---|---|---|---|---|---|---|
| Total no. of observations | 8820 | 1688 | 1034 | 654 | 1034 | 1688 | 1034 | 654 | 1034 |
| Total no. of TILs | 1589 | 290 | 215 | 119 | 99 | 280 | 208 | 190 | 188 |
| Percent of TILs (%) | 18.0 | 17.2 | 20.8 | 18.2 | 9.6 | 16.6 | 20.1 | 29.1 | 18.2 |

When analyzing the number of occurrences and the rate of TIL from 1995 to 2016 (Table 2), the total number of surveys was 8820, with 1589 TILs and a 18.0% occurrence rate. The most frequent survey was 1688 in February and August, 1034 in April, June, October, and December, and 654 in May and November. The incidence of TILs was high in February and August, 290 and 280, respectively (the incidence was 17.2% and 16.6%, respectively). However, the probability of occurrence was 29.1%, which was the highest in November, due to the fact that the number of surveys was relatively less than that in February and August, and only the southern waters of Jeju Island were surveyed. The probability of a TIL in June was the lowest at 9.6% (this season is considered to have the least impact on YSCW and TWC).

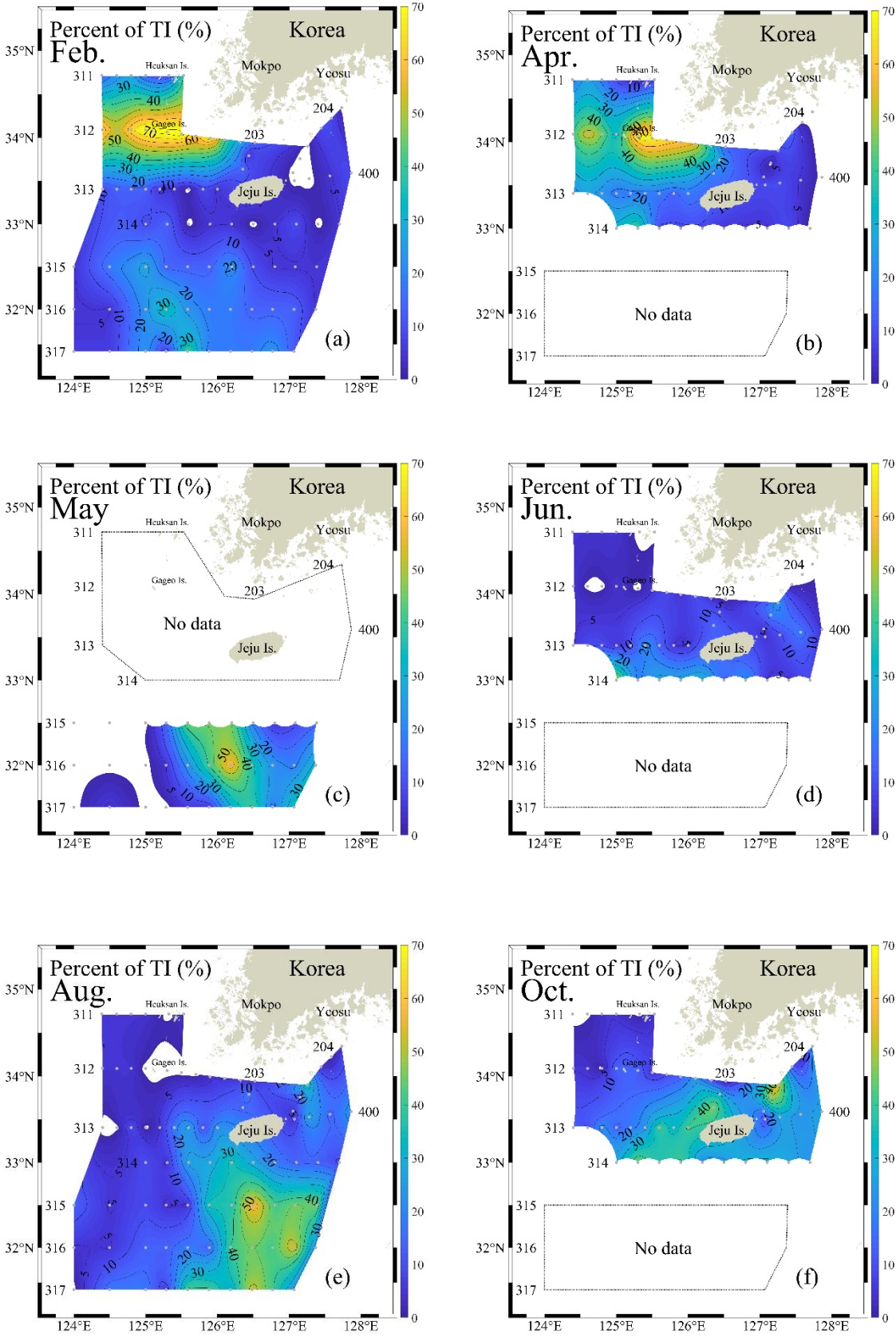

**Figure 3.** *Cont.*

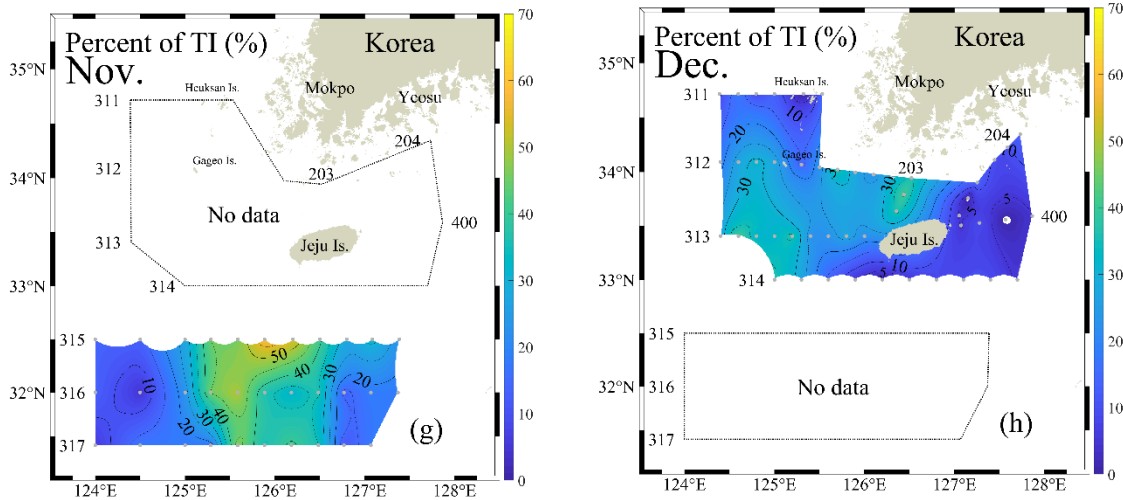

**Figure 3.** Percent of occurrences of water temperature inversion layer in the North East China Sea during 1995 to 2016 in February (**a**), April (**b**), May (**c**), June (**d**), August (**e**), October (**f**), November (**g**), and December (**h**).

### 3.2. T–S Diagram Analysis of TIL

The water masses affecting the surrounding waters of NECS include YSCW, TWC, and JWC. The water temperature and salinity of the TIL distribution in the study area with various water masses were analyzed using a T–S diagram, and the characteristics of seasonal water masses in February and August were analyzed (Figure 4). In the T–S diagram, YSCW has a range of water temperatures of 5.0–14.5 °C, with salinity of 32.0–33.5 psu, JWC has a water temperature of 12.0 °C or higher and salinity of 33.5–34.0 psu, and TWC has water temperature of 14.0 °C or higher and salinity of 34.0 psu or higher (Table 1). In the T–S diagram in February, three water masses were in the TIL. It is mostly present in the low water temperature range. The distribution of YSCW (blue dot) appears to be the most widely distributed. Many water masses not included in the three water mass ranges are also distributed. In the TS diagram of August, it was distributed variously in the TIL. It has reached high water temperature ranges and low salinity ranges. It was also within the range of three masses. However, in the T–S diagram (Figure 4), it was not enough to determine the quantitative distribution of each water mass.

The proportions of water masses present in the TILs were seasonally different, and the three water masses had different seasonal intensities (Figure 5). In February, the proportion of the water mass in TIL was 4.8% for YSCW, 0.7% for JWC, and 2.6% for TWC. We believe that this is due to the 311 transect and 312 transect, which had the highest probability of TIL in February. In August, there was a high probability of TIL occurring in the south and the surrounding areas of Jeju, which resulted in YSCW 0.7%, JWC 2.3%, and TWC 1.4%. In the winter season (February), YSCW is the leading cause of TIL, and in the summer (August), JWC and TWC are the leading cause. TILs are influenced by seasonally and spatially different water masses.

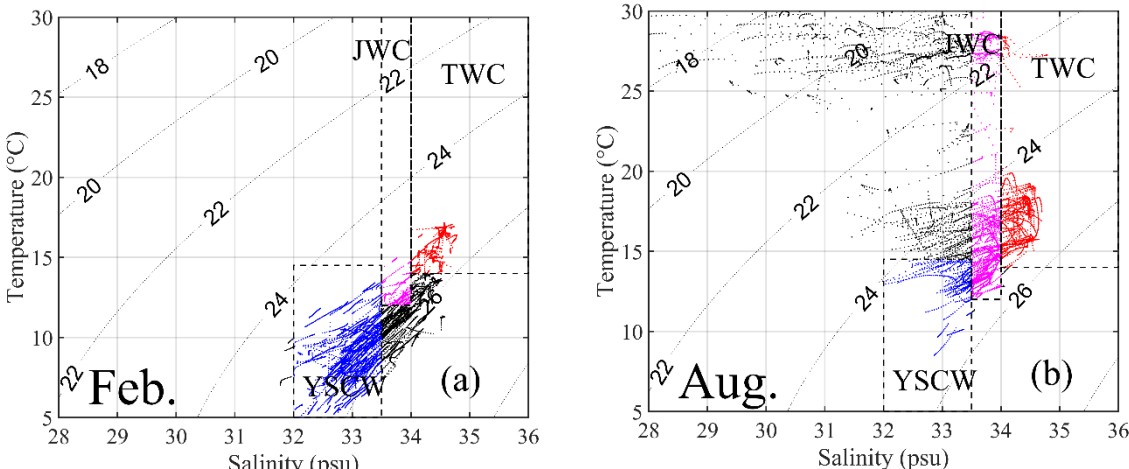

**Figure 4.** The distribution of water temperature and salinity in the water temperature inversion layer in February (**a**) and August (**b**). JWC is Jeju Warm Current (magenta dot), TWC is Tsushima Warm Current (red dot), and YSCW is Yellow Sea Cold Water (blue dot). The broken lines indicate the ranges of the different water masses.

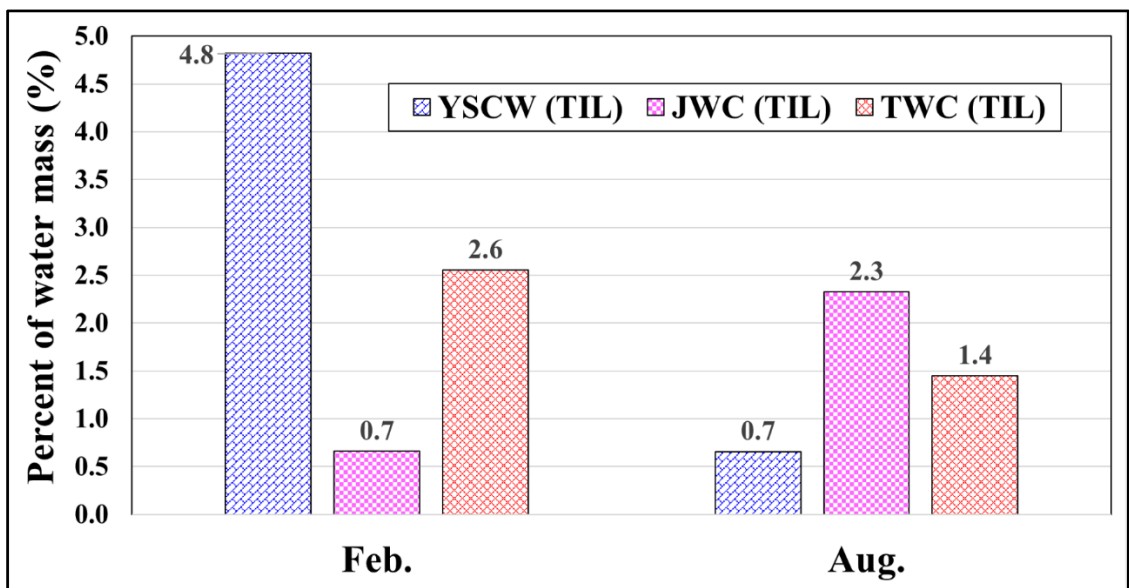

**Figure 5.** Percent of water mass of Yellow Sea Cold Water (YSCW), Jeju Warm Current (JWC), and Tsushima Warm Current (TWC) in water temperature inversion layer in February and August from 1995 to 2016.

## 4. Discussion

### 4.1. Comparison of Winter and Summer Characteristics of TIL

The probability of TIL occurrence differed between winter and summer depending on the observation transect (Table 3). The most likely cases of winter occurrence were two transects out of 10 transects, 311 transect and 312 transect, and 25 times higher in winter (62.6% in February and 2.5% in August) near Gageo Island (312 transect). In the Huksan Island (311 transect), winter was 5.6 times higher, with 21.2% in February and 3.8% in August. The high probability of summer occurrence was found in eight transects out of 10 transects (203, 204, 400, 313, 314, 315, 316, and 317 transect). In the case of the 203 transect, summer was 1.1 times higher, with 10.6% in February and 12.1% in August. The 204 transect, the summer was 3.5 times higher, with in February 3.0 % and 10.6 % in August, while

the 400 transect was 5.0 times higher in summer at 3.4 % in February and 17.0% in August. For 313 transect, the summer was 1.2 times higher at 8.1% in February and 10.1% in August, while the summer at 314 transect, the summer was 3.7 times higher at 5.0% in February and 18.6% in August. In the 315, 316, and 317 transects, February was 12.4%, 15.7%, 17.1%, August was 27.3%, 24.8%, and 27.1%, and summer was 2.2, 1.6, and 1.6 times higher than winter.

**Table 3.** The percent of TIL for the observation transect in February and August.

| Transect / Month | 203 | 204 | 400 | 311 | 312 | 313 | 314 | 315 | 316 | 317 | Mean | STD |
|---|---|---|---|---|---|---|---|---|---|---|---|---|
| Feb. | 10.6 | 3.0 | 3.4 | 21.2 | 62.6 | 8.1 | 5.0 | 12.4 | 15.7 | 17.1 | 15.9 | 17.5 |
| Aug. | 12.1 | 10.6 | 17.0 | 3.8 | 2.5 | 10.1 | 18.6 | 27.3 | 24.8 | 27.1 | 15.4 | 9.1 |
| Times | 1.1 | 3.5 | 5.0 | 5.6 | 25.0 | 1.2 | 3.7 | 2.2 | 1.6 | 1.6 | 1.0 | 7.1 |

　　When winter (Feb.) is high.　　　　When summer (Aug.) is high. STD is standard deviation.

TILs in the Yellow Sea and South Sea of Korea were reported to occur in winter. The study in [3] analyzed the TILs that occurred in the West Sea of Korea between 1965 and 1979 and found that February (winter) is seven times higher than August (summer). The work in [7] reported that, in the South Sea of Korea, TIL occurred six times more frequently in winter than in summer during 1965 to 1979. In this study, however, the probability of occurrence in February and August was similar, with 15.9% and 15.4%, respectively. Since the study area includes the south of Jeju Island, which is heavily influenced by JWC and TWC, the probability of summer occurrence was higher than that of other studies. In the study in [6] on the summer TIL on the east coast of Jeju Island, TIL occurred in 37 out of 67 stations (58.2%). The work in [3] also found that the probability of TIL incidence increased by more than 40–70% in the western sea of Jeju Island in the summer. In this study, the difference in the probability of the occurrence of water temperature inversion layer in winter was not significant, but the difference in seasonal location (Gageo Island, south of Jeju Island) was confirmed.

*4.2. Current Characteristic*

The GLORYS12 model of Global Ocean Physics Reanalysis was used to investigate the current characteristics in the study area. February (a) and August (b) in the 2015 data were used to examine the monthly average current characteristics (Figure 6). The dotted transect in the figure is the study area. In February, the current speed of JWC and TWC is about 25 cm/s at the depth 29.4 m. In August, the current speed of JWC and TWC were 30–50 cm/s at the depth 29.4 m, more than double that in February. The current extends strongly from the South of Jeju Island to the South Sea of Korea. Based on the average value of the northward component between 1995 and 2016 at latitudes of 31° N and longitudes of 123 to 128° E, the inflow into the study area was calculated (Figure 7). The northward component at 123.0–124.0° E was judged to be due to the Taiwan Current Warm Water (TCWW) of northward velocity being less than or equal to 10 cm/s, and the strong northward component at 126.0 to 128.0° E is due to the influence of TWC of northward velocity being more than 15 cm/s (maximum is 18 cm/s).

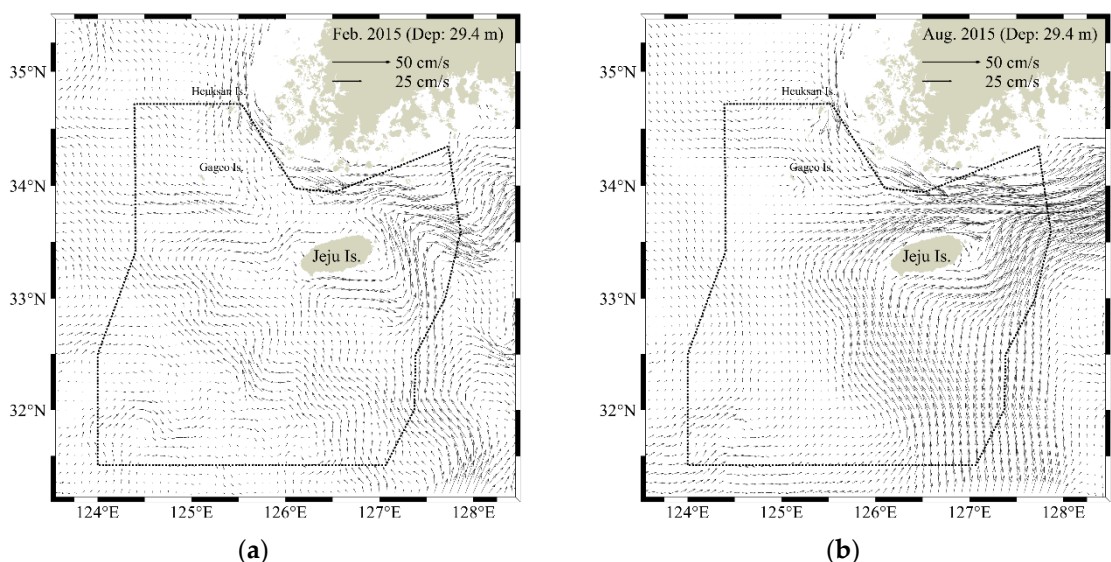

**Figure 6.** Global Ocean Physics Reanalysis GLORS12 model results: Current velocity in February (**a**) and August (**b**) in 2015 (the dotted line is the study area).

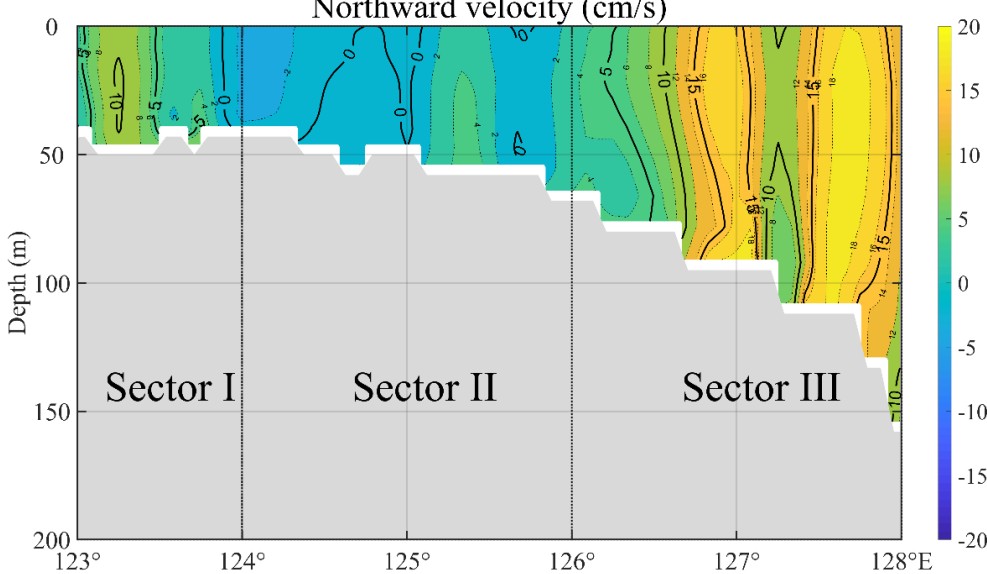

**Figure 7.** Vertical distribution of the average northward velocity from 1995 to 2016.

For the volume transport in the study area, data from 31° N, 123° E to 128° E of the GLORYS12 model were used. The volume transport was analyzed for the monthly and annual average transport from 1995 to 2016 (Figure 8a). The dotted line, which is the average value for each year, shows a decreasing trend for 1998, 2014, and 2015. The tendency of annual fluctuations; however, was difficult to determine. The monthly mean values were also calculated (Figure 8b). The monthly volume transport started at 2.59 Sv in January and rose to 4.77 Sv by July. It then decreased from 4.74 Sv in August to 3.01 Sv in December. The volume transport is shown, and calculations were made for each sea area identified in Figure 7. The volume transport by sector was 9.6% (an average of 0.35 Sv) in Sector I, 8.1% (an average of 0.30 Sv) in Sector II, and 82.3% (an average of 3.05 Sv) in Sector III. Most of the currents that flow north at 31° N were introduced into the study area through Sector III.

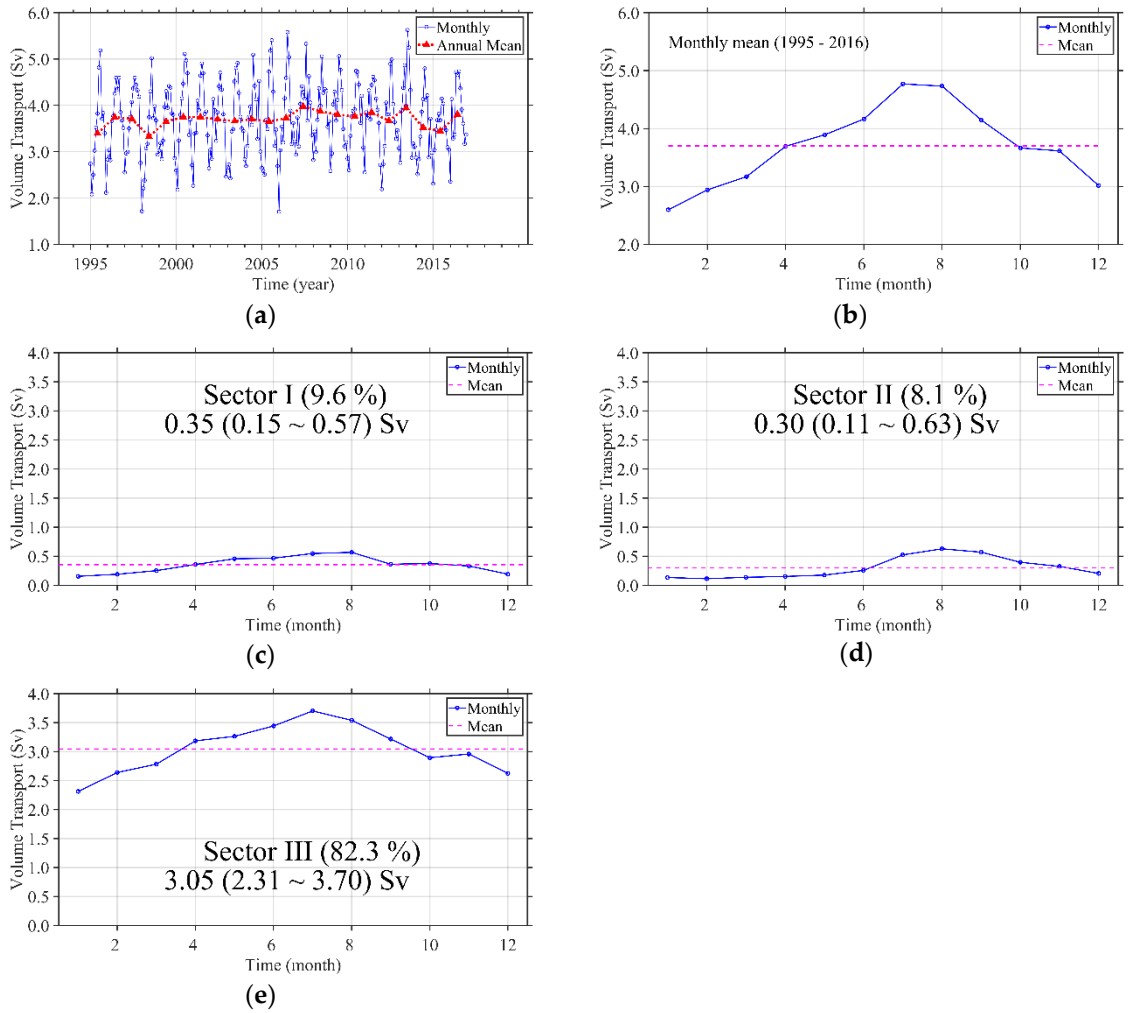

**Figure 8.** Annual distribution of volume transport into the study area from 1995 to 2016 (**a**). Monthly average distribution of volume transport (**b**). Monthly average distribution of volume transport for sector I (**c**), sector II (**d**), and sector III (**e**).

*4.3. Correlation Analysis of Monthly Occurrence Characteristics of TIL with Volume Transport*

TIL occurs mainly at the thermohaline frontal area where different water masses meet [9]. We analyzed the annual and seasonal changes of volume transport of the study area to find the causes of TIL in the study area. The spatial and temporal results of annual and seasonal TILs were identified at each transect of the study area. The correlation between TWC and TIL, which is considered to be the dominant current in the study area, was confirmed (Figure 9). The correlation between TIL monthly occurrence probability of each station and calculated correlation coefficient of monthly volume transport at Sector III was confirmed. A correlation coefficient of 0.8 or higher was found in the southern waters of Jeju Island, and a correlation coefficient of −0.6 was found in the 312 transect near Gageo Island.

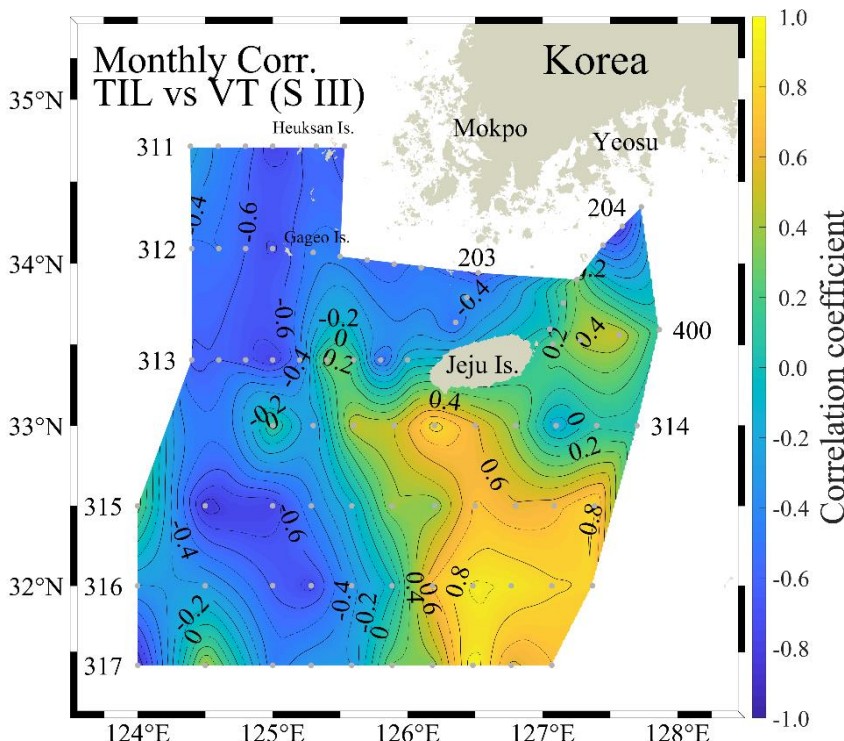

**Figure 9.** The distribution of the correlation coefficient for the monthly mean of the Sector III volume transport and the percentage of TIL at all stations.

As a result of the T–S diagram analysis, most of the incoming seawater from the south was analyzed to TWC, that has high-temperature, high-salinity water and originates from Kuroshio. Therefore, the probability of TIL occurrence in the south of Jeju Island will be highly correlated with the strength of the large manna stream flowing into the study area. The most influential factor in TIL generation in the southern waters of Jeju Island is the seasonal change in the amount of TWC flowing into the study area. The front is formed by various water masses flowing from the south of Jeju Island, thus the probability of TIL occurring mainly from the front is increased.

According to the authors in [3], TIL occurs when YSCW and YSWC, which differ in water temperature and salinity, form a boundary in the western of Jeju Island and are mixed with each other. In the study [6], who studied east of Jeju Island, analyzed that TIL occurs by the interaction intrusion between KCC and TWC water masses. Furthermore, study [9] reported that TIL occurs when YSCW of low-temperature and low-salinity water in the middle layer meets JWC of high-temperature and high-salinity water in the lower layer in winter. In this study, we analyzed the monthly variation of volume transport for TWC affecting TIL generated in the NECS, and suggested correlation with spatial occurrence probability of TIL. Therefore, TIL generated from NECS is formed by TWC flowing south of Jeju Island in the summer, and TIL in the area near Gageo Island is more closely related to YSCW behavior than TWC influx.

## 5. Conclusions

To investigate the behavioral characteristics of TIL in NECS, 8820 survey data were analyzed from 1995 to 2016. Among them, TIL appeared 1589 times and the probability of occurrence was 18.0%. In February, it was the highest at 62.6% on the 312 transect near Gageo Island. TIL at 312 transect near Gageo Island is 25 times higher in winter than in summer. In August, the highest occurrence was 27.3% at 315 transect in the southern of Jeju Island. In the 314 transect south of Jeju Island, summer was 3.7 times higher than winter. The probability of TIL occurrence by seasons differed by temporal and spatial. In the study [3] found that probability of TIL occurrence in summer at the yellow sea is

seven times higher than winter. However, in the southern of Jeju Island, the probability of TIL was higher in summer than in winter.

As a result of analysis of T–S diagram in TIL, YSCW was found to be high in winter, JWC and TWC were high in summer, and it was possible to distinguish the water mass causing seasonal TIL. Monthly variability of volume transport into NECS increased from January to July and peaked in August, and decreased from September to December. The volume transport inflow was divided into three regions, and Sector III, which accounts for 82.3% of total transport volume, was considered as the region of the TWC influx. As a result of correlation analysis of monthly TIL occurrence rate volume transport from Sector III, the correlation coefficient was over 0.8 in the southern Jeju Island. On the contrary, the correlation coefficient was around −0.6 for Gageo Island near the 312 transect. The temporal and spatial correlation between volume transport and TIL is presented.

The water temperature inversion layer changes the vertical structure of the water temperature, and the underwater sound propagation characteristics are changed according to the shape of the water temperature inversion layer. In the future, it is necessary to study the underwater sound propagation pattern according to the water temperature inversion layer.

**Author Contributions:** Writing—original draft, S.H.K.; Conceptualization, B.K.C.; Formal analysis, E.K. All authors have read and agreed to the published version of the manuscript.

**Funding:** This research was a part of the project titled "Development of technology on survivor search and survivability assurance in the accident ship", funded by Korea Coast Guard, Korea (Grant No. PN68020). This work was also a part of the project titled "Construction of ocean research stations and their application studies", funded by the Ministry of Oceans and Fisheries, Korea (Grant No. PM61100).

**Acknowledgments:** Thanks are due to Byoung-Nam Kim, Ho youn Ji, Mi Ran Kim, Min Gu Kim and Suin Lee for their valuable guidance and discussion.

**Conflicts of Interest:** The authors declare no conflict of interest.

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
