# Peer review of "Study on the Behavior of the Water Temperature Inversion Layer in the Northern East China Sea"

_jmse, doi:10.3390/jmse8030157_

Round 1

Reviewer 1 Report

The article titled "Study on behaviour of water temperature inversion 2 layer in the Northern East China Sea" deals with the interesting issue of water temperature inversion occurring in the northern part of the East China Sea. The authors used data on water temperature and its salinity from eleven measuring profiles in the period 1995-2016.

Despite the use of many valuable data, the article was written very carelessly. The study contains a lot of errors that make the presented text in many places incomprehensible. Examples of rows: 14-15, 45, 97, 106, 107, 132, 133 and others. Authors did not list units describing quantitative relationships, e.g. row 67. Literally, literature is unequal, e.g. row 283-284. The text has unremoved fragments after a likely correction, e.g. row 193, 257. In addition, rows 345 to 356 in the reference list should be deleted. It can be found other typing errors, e.g. in table 3 the abbreviation Feg. should be replaced by Feb. The description of Figure 9 was repeated in rows 275 and 276.

In addition to the carelessly written text, the authors should pay attention to the following issues:

1 - abstract should contain a description of the most important results. It should be clearly written, taking into account the outline of the used research methods.

2 - The description of the research methods should also be written clearly with a detailed description, e.g. depth of water temperature measurements and their time interval.

3 - The interesting issue of the occurrence of water temperature inversions in the investigated basin should also be clearly described using available scientific literature.

Although the reviewed article concerns an interesting issue in its current form, it is not suitable for publication.

To make the text suitable for publication, the authors should certainly devote more attention to the scientific and editorial correctness of the presented issues.

Author Response

The request has been modified. Thank you so much for your review.

Reviewer 2 Report

see attached file

Author Response

(The authors gave the same response as above.)

Round 2

Reviewer 1 Report

The authors made corrections in line with the reviewer's previous suggestions.
The text of the article requires only a small correction:
1 - The description of Figure 4 does not explain the colors used
2 - line 270 - it is necessary to delete the crossed out "the".
After corrections have been made, the text is printable.

Author Response

I think it could be a better article. Thank you very much for your comment.

Reviewer 2 Report

The authors answer to some of my requests (just the lists that I included), but they did not reach the core of my suggestions. The paper still has several deficiencies: no indication about uncertainties, no adequate description of the analytical methods, no description of the data analysis.

In my opinion, this paper is just enough for publication; I think that the authors would have produced a very good paper with some more efforts, but I feel like they are more interested in publishing than realizing a good work.

In synthesis, I think that this paper can be published with a minor revision which includes at least some reference on the importance of uncertainty analysis, in-depth description of the methods and data analysis.

Author Response

(The authors gave the same response as above.)
